# Formulation of Artificial Diets for Mass-Rearing *Eldana saccharina* Walker (Lepidoptera: Pyralidae) Using the Carcass Milling Technique

**DOI:** 10.3390/insects13040316

**Published:** 2022-03-23

**Authors:** Nomalizo C. Ngomane, Elsje Pieterse, Michael J. Woods, Des E. Conlong

**Affiliations:** 1Department of Conservation Ecology and Entomology, Stellenbosch University, Stellenbosch 7600, Western Cape, South Africa; nomalizo@riverbio.com; 2South African Sugarcane Research Institute, 170 Flanders Drive, Mount Edgecombe 4300, KwaZulu-Natal, South Africa; 3Department of Animal Science, Stellenbosch University, Stellenbosch 7600, Western Cape, South Africa; elsjep@sun.ac.za (E.P.); michael@susento.com (M.J.W.)

**Keywords:** comparative slaughter technique, insect rearing, sterile insect technique, sugarcane, mass production

## Abstract

**Simple Summary:**

Human food needs have driven the formulation of scientifically derived artificial diets for large scale sustainable production of fish, birds and animals. The carcass milling technique (CMT) has been instrumental in its achievement. It formulates specific diets from nutrient analyses of the species to be mass-reared, and their natural host foods. These diets do not deleteriously impact growth or development, sometimes enhancing this. In recent times, the CMT is being used to formulate artificial insect diets to provide high quality individuals needed to feed other organisms, and treat large areas of agricultural land containing crops and animals in sustainable area wide-integrated pest management (IPM) programmes. CMT was used to formulate four diets to mass produce *Eldana saccharina* needed for a sterile insect program, in a cost-effective nutritious way without compromising fitness. Yields in terms of pupae obtained from the larvae inoculated were high for all diets (97% at day 27), as was female fecundity and fertility (>870 eggs per female; >90% fertile). Quickest larval development time (17% pupae at day 20), highest adult emergence (98%), and females mated with most males (mean = 3) occurred from the individuals reared on the minimum specifications (MS) CMT diet, making it the preferred choice to replace the routinely used diet.

**Abstract:**

The carcass milling technique (CMT) formulates specific diets from nutrient analyses of species to be reared and their natural host plants. The first of four diets developed used the minimum ingredient specifications (MS) of published diets for *Eldana saccharina*. The remaining were based on the ideal amino acid composition and profile (IAAP) of its second (IAAP2), third/fourth (IAAP3/4) and fifth/sixth (IAAP5/6) instar larvae. The control was a modified *Ostrinia nubilalis* diet. Survival to pupae of inoculated *E saccharina* neonates was high on all CMT formulations (>92% at day 20 and >97% at day 27). Larvae developed fastest on the IAAP3/4 and MS diets (25% and 17% prepupae and pupae on day 20, respectively). Pupal weights were not significantly influenced by CMT diets (0.1121 g male; 0.1864 g female). The control group produced heavier male and female pupae (0.1204 g; 0.2085 g, respectively). Adult emergence was highest from the MS (98%), then the IAAP3/4 (97%) and control (96%) diets. Sex ratio of adults from all diets was close to one. Males from the IAAP5/6 diet mated with significantly more females (six), and females from the MS diet mated with more males (three) than those from remaining formulations. All females produced in excess of 870 eggs, more than 90% were fertile after mating. The pH (4.79); moisture content (81.43%) and water activity (0.92 a_w_) of the diets were not significantly different, maintaining quality and stability throughout the larval period, ensuring optimal growth and development. The MS diet formulation was the preferred choice to replace the current *E. saccharina* diet.

## 1. Introduction

Mass-rearing of insects has expanded dramatically in the agricultural industry for the development and support of integrated pest management (IPM) approaches [1]. Books have been written outlining specific diets for different insect species [2], on how to develop insect diets [3] and outlining the principles and procedures for rearing high-quality insects [4]. In the local context, from as early as the 1980s, papers from the South African Sugarcane Research Institute (SASRI) were published on rearing insects on artificial diets for research into host plant resistance, push–pull technology, biological control and sterile insect technique (SIT) programmes [5,6,7,8,9,10].

Most artificial diets and/or feeds are composed of a mixture of nutrients enabling animals to maintain their high life-cycle vigour [11,12]. Insects consume approximately 70 to 75% of their diet to provide for their life stage maintenance, and as diet expense is relatively higher than insect production costs, it is essential to avoid an over-supply of nutrients, because once the insect’s nutrient requirements are supplied, any excess nutrients are excreted or stored as unwanted fat by the insect, and are thus wasted [11,12]. Oversupply of nutrients also contributes to the build-up of primary or secondary metabolites which may be toxic, antagonistic or result in imbalances that lead to increased metabolic stress in the insect. Furthermore, undersupply or the absence of nutrients for the reared insects may lead to a total breakdown in production, whereas minimal supply results in immune suppression, reduced productivity or a reduction in fecundity and fertility [13]. However, to minimise these shortfalls in diet production, several techniques have been developed to evaluate actual nutritional requirements of insects to be fed, to help better formulate insect diets [13].

The carcass milling technique (CMT), a version of the comparative slaughter technique, plays an important role in the development of animal feeds and, most recently, insect diets [13]. This technique requires that representative animals or insects (and their natural food) are slaughtered and analysed for dry matter, crude protein and crude fat, using proximate and amino acid analyses [14]. The technique has been successfully used to determine baseline nutrient specifications for mass rearing the false codling moth *Thaumatotibia leucotreta* Meyrick (Lepidoptera: Tortricidae) and the black soldier fly *Hermetia illucens* L. (Diptera: Stratiomyidae) [15,16]. Using information collected from the proximate and amino acid analyses, relatively inexpensive artificial diets for insects have been formulated using Windows-based feed formulation programmes [13].

Several artificial diets have been developed for mass rearing the African sugarcane stalk borer, *Eldana saccharina* Walker (Lepidoptera: Pyralidae). All diets developed effectively supported optimal survival and development of *E. saccharina*. For example, Atkinson [17] developed the first diet to provide material for biological studies of the insect. As the biological control programme against *E. saccharina* gained momentum, higher numbers of this pest had to be reared to provide hosts for the respective biocontrol agents and material for plant resistance trials [18,19]. A complete artificial diet for bioassay purposes was developed [20,21] and more recently, the diet was further refined for SIT trials [9,10].

The diet of Ngomane et al. [10] was based on a diet developed for the European corn borer *Ostrinia nubilalis* Hubner (Lepidoptera: Crambidae) [22]. This diet was formulated based on ingredient composition and not on the nutrient requirements of growing *E. saccharina* [13]. The current study, therefore, provides a different approach to developing *E. saccharina* diets, based on animal science principles using the CMT, and body and chemical composition studies. The aim of this technique in diet production is to formulate the chemical composition of these diets unique to the particular insect being reared, to provide the optimum balance between carbohydrates, proteins (including amino acids), lipids, vitamins and minerals, in addition to specific requirements for the sterols needed by that specific insect.

The specific objectives of this study were to formulate a new artificial diet based on the CMT and then compare growth and fitness parameters of *E. saccharina* reared on this diet and that of the diet developed by Ngomane et al. [10], which served as the control diet for this study.

## 2. Materials and Methods

### 2.1. Experimental Site

The study was conducted at the SASRI Insect Rearing Unit (IRU), Mount Edgecombe, KwaZulu-Natal, South Africa. The rearing facility provides reliable temperature and humidity control, and light through windows supplemented by T12 65 W cool white fluorescent tubes. Laboratory standard operational procedures developed for the SASRI-IRU were followed to prevent contamination and contamination spread in the diets being tested [10]. All temperature (°C) and relative humidity (RH) conditions were kept similar for all aspects tested for both the formulated and control diets (26 ± 2 °C, 72 ± 5% RH). However, the photoperiod was different for rearing (0 Light (L): 24-h Dark (D)), quality (8-h L: 16-h D), and oviposition assessments (8-h L: 16-h D). The 0 L: 24-h D photoperiod was used as larvae were at the life stage of being fed on the diets, and these are generally feeding inside stalks, thus the light could not reach them. On the other hand, the oviposition trials involved adults that are exposed to daylight in their natural life cycle, hence the 8-h L: 16-h D photoperiod.

### 2.2. Carcass Milling Technique (CMT)

#### 2.2.1. Collection and Treatment of *Eldana saccharina* Larvae

*Eldana saccharina* larvae were routinely reared on the Ngomane et al. [10] diet, in the SASRI-IRU larval growth rooms, harvested by hand from diet trays (inoculated 15 to 20 days before) and separated into different larval instar groups (i.e., 2nd instar, 3rd/4th instar, &5th/6th instar). The collected 2nd instar larvae were transferred into boiling water (100 °C) for 1 min, and the 3rd/4th and 5th/6th instar larvae for 1.5 min. These exposure times and water temperatures prevented further metabolic activity, and thus further denaturing of their larval proteins [13]. Underexposure in terms of time and water temperature would allow the larval normal metabolic activity to continue, thus giving an erroneous protein reading [13]. Thereafter, larvae were cooled rapidly and stored in a Bio Compact freezer II 410 (Gram Commercial, Vojens, Denmark) at −20 °C. A sample of 50 g of larvae per instar group was required for the proximate and amino acid analysis. This was packed separately into clean plastic vials, lids screwed on tightly, sealed in a clear plastic sleeve, labelled accordingly (i.e., larval instar group) and kept in a polystyrene cooler box (Algoa Plastics, Port Elizabeth, South Africa) packed with three frozen ‘Seagull’ solid, jumbo ice bricks (Seagull Industries, Cape Town, South Africa). The cooler box and contents were couriered overnight to the Department of Animal Sciences, Stellenbosch University, Western Cape, South Africa for proximate and amino acid analyses to be completed.

#### 2.2.2. Collection of Natural Host Plants of *Eldana saccharina*

Four 12-month-old sugarcane stalks (variety NCo 376), were collected from SASRI Field number 14 (29°42′15.25″ S 31°02′40.93″ E) and the bottom section was cut into short pieces of the pith (with the rind of the sugarcane stalk removed) to make up 200 g of fresh samples. Mature, fully expanded papyrus (*Cyperus papyrus* L.) umbel meristems, comprising the top 2 cm of the culm, with all fronds dissected off it (to make up 200 g of fresh samples) were collected on the SASRI property/waterways. These are the plant portions (natural diets) most fed on by *E. saccharina* larvae. The collected host plant materials were sealed separately in clear plastic sleeve bags just larger than the fresh plant material collected, labelled accordingly and stored in the Bio Compact freezer at −20 °C. The plant samples were sent separately in polystyrene cooler boxes each packed with three frozen ‘Seagull’ solid jumbo ice bricks. The cooler boxes and contents were couriered overnight to the Department of Animal Sciences, Stellenbosch University for proximate and amino acid analyses to be completed.

#### 2.2.3. Proximate and Amino Acid Analyses

Thirty sub-samples of the larval carcasses and natural diets of the larvae were dried in a Labcon FSOH 16 Oven (CC Imelmann, Johannesburg, South Africa) at 60 °C for 24 h. They were each homogenised using a KN 195 Knifetec^™^ mill (FOSS, Hillerød, Denmark) and subsamples were taken to determine their IAAP Profiles [13].

#### 2.2.4. Amino Acid Analyses

The process involved weighing 0.1 g of a sample, placing it in a specialised hydrolysis tube and adding 6 mL of hydrochloric acid and 150 mL of phenol [13]. The tubes were vacuated and nitrogen added under pressure. These were sealed off with a blue flame and samples left for 24 h to hydrolyse at 110 °C [13]. Thereafter, the samples were transferred to 0.5 mL Eppendorf microcentrifuge tubes and kept in the refrigerator at 4 °C until they were ready to be sent to the Central Analytical Facility of Stellenbosch University, where amino acid composition was determined by means of the water AccQ Tag Ultra Derivation method [13].

#### 2.2.5. Proximate Analyses

The moisture, crude protein and ash of the natural diets and the different larval instar carcass groups were analysed according to the methodology of the Association of Official Analytical Chemists [13]. To determine the moisture content, 2.5 g of the homogenised sample was weighed, dried in an oven for 24 h at 100–105 °C and then re-weighed. The dried sample was incinerated in a muffle furnace (Thermo Fisher Scientific, New York, NY, USA) for 6 h at 500 °C. It was then cooled using a Glass Vacuum Desiccator 8” (210 mm) (Deschem Science Supply, USA) and re-weighed to provide an estimate of the ash content [13]. To determine the total lipid content of the samples, a rapid solvent extraction method was used [23]. A ratio of 1:2 chloroform: methanol solution was used for the plant samples due to their relatively low-fat content (<50 g kg^−1^, USDA, 2019). A ratio of 2:1 chloroform: methanol solution was used for the larval carcass samples, as initial tests showed a lipid content of more than 50 g kg^−1^ [13]. To determine the total protein content, the LECO combustion method was used [13]. The dried defatted samples were crushed using a 120 W Multi-purpose Coffee Grinder, and for each sample an aliquot of 0.5 g was weighed into a LECO foil cup (Thermo Fisher Scientific, New York, NY, USA), which was incinerated and then analysed for nitrogen content using EDTA for calibration. To obtain the protein content, the nitrogen content was multiplied by 6.25 [13].

### 2.3. Artificial Diet Formulation Using WinFeed

Six artificial diets, namely minimum specification (MS); ideal amino acid composition and profile of the 2nd instar larvae (IAAP2); ideal amino acid composition and profile of the 3rd/4th instar larvae (IAAP3/4); ideal amino acid composition and profile of the 5th/6th instar larvae (IAAP5/6); sugarcane (SC) and papyrus (PAP) diets, were formulated from the proximate and amino acid analyses using WinFeed 3.0 (a Windows-based feed formulation programme developed by EFG Software. See http://www.winfeed.com, accessed on 9 March 2022) [13]. The MS diet was formulated using the minimum amounts of common dietary ingredients obtained from a summary of published *E. saccharina* diets [9,10,17,18,19], on which the insect was previously successfully reared. The IAAP2, IAAP3/4 and IAAP5/6 diets were formulated using ideal amino acid compositions and profiles, determined from the CMT, which resembled the amino acid composition and profiles of the different larval instar groups. The SC and PAP diets were formulated using the nutrient compositions, determined by proximate and amino acid analyses, of the natural diets of *E. saccharina* [13]. The control diet was reverted to nutrient composition using WinFeed and the calculated nutrient composition was used as the control nutrient specifications. The experimental diets were formulated to unique nutrient specifications using similar ingredients to that of the control diet [13].

To formulate the MS diet, raw ingredients present in published *E. saccharina* diets were entered into the “ingredient” tab of WinFeed. The cost (South African Rands) per ton and bag weights (g) of the ingredients (although not essential for this study) were also entered. All essential nutrients available in the ingredients were selected on the “nutrient” tab. The nutrient composition of each of the ingredients were obtained from various internet sources i.e., Feedipedia, the Food and Agriculture Organization of the United Nations, Google Scholar and Google search. The amino acids and minerals entered into WinFeed were expressed as a percentage of protein sample and ash sample, respectively, on a dry matter basis. The dry matter percentage was entered in its present state. In the “animals” tab, the experimental insect, *E. saccharina* was entered. All ingredients and nutrients were made applicable to the specified insect. Compositions of the diets found in the literature were created in the “compositions” tab. For each diet composition, ingredients were selected, and their weights were entered. The diets were then brought to a scale of 100%.

Once all the diet compositions were entered into WinFeed, a summary report in Excel, showing the compositions with their associated ingredients and ingredient prices, could be accessed. At the bottom of the table, the costs of the compositions were calculated. In the same Excel file, a nutrients table showed all the available nutrients found in the diet compositions entered. Using this table, the minimum specification values of the nutrients were calculated. Back on WinFeed, a feed was created. The minimum specification values were entered on the “feed” tab under “minimum”. Applicable nutrients and ingredients were selected. Once all the information was entered, the diets were formulated [13] (Table 1).

The formulated diets contained lucerne meal and the control diet contained rabbit meal (Mkondeni Animal Feeds CC, Pietermaritzburg, South Africa) as their main ingredients. These were crushed into powder, using a SNU506-605 Baby Hippo Hammer Mill (Collins and Son, Umhlanga Rocks, South Africa) and sterilised in a Memmert UL80 force draft oven (Gemini BV, Apeldoorn, The Netherlands) at 65 °C for 2 days. The control diet used agar as a gelling agent and the formulated diets used carrageenan gel as a cheaper alternative to agar [13]. The control diet included yeast extract and the PAP diet included L-lysine HCL, as an added amino acid source. The SC, PAP and control diets also included sodium chloride, which is critical for osmoregulation, neuromuscular mechanisms, digestive and excretory processes in insects [24]. The remaining diet components commonly included carbohydrates and proteins (i.e., egg powder, chickpea flour, wheat bran, full cream milk powder and sucrose), vitamins and minerals (i.e., vitamin premix), pH modifiers (i.e., tri-sodium citrate and citric acid) and preservatives and antimicrobial agents (i.e., ascorbic acid, nipagin, sodium propionate, acetic acid and oxytetracycline).

### 2.4. Eldana saccharina Rearing Process

#### 2.4.1. Diet Preparation

Diet ingredients were weighed, using a calibrated two decimal place Mettler Toledo ML6001 New Classic MF balance (Microsep (Pty) Ltd., Johannesburg, South Africa), according to the formulated diet recipes (Table 1) in the IRU diet kitchen, maintained at 22 ± 2 °C and ambient humidity. The weighed dry content ingredients of each formulated diet was poured into the bowl of a 6.7 L Kenwood Titanium Major KMM060 food mixer and thoroughly mixed for 1 min. Boiling water at a ratio of 1:3 (500 g:1500 mL dry matter to water), together with 8 mL of acetic acid (dissolved in water), was added to the mixture. The food mixer was allowed to run for another minute. The resulting mixture was poured into a 2.5 L plastic mixing bowl and placed in a 700 W microwave oven for 2 min on high heat to cook. Extra care was taken to ensure that the diet did not overcook. As soon as the diet started bubbling within the 2 min, it was removed from the microwave, thoroughly mixed using a sterilised spoon (dipped in Denol (70%) and then in distilled water) and then placed back in the microwave to continue cooking until the 2 min cooking period was completed.

The preparation of the control diet followed the same procedures, with the exception that the agar solution was prepared separately. In a 1 L plastic jug, 500 mL of boiling water was dispensed into which 4.6 g of agar powder was poured slowly and stirred using a spoon to avoid the formation of agar lumps. The agar solution was poured into a 1 L Schott Pyrex bottle and autoclaved at 121 °C for one hour. Once autoclaved, the hot agar was poured into the running food mixer and a balance of 1000 mL of boiling water with 8 mL of acetic acid was added to the autoclaved agar solution, followed by the weighed dry mixture. The resulting mixture was poured into a 2.5 L plastic mixing bowl and placed in a microwave oven to cook for 2 min on high heat.

#### 2.4.2. Diet Dispensing

Seven hundred 25 mL plastic screw top vials, for each formulation described in Table 1, were sterilised overnight in a 0.5% sodium hypochlorite (NaOCl) solution. Upon removal, they were shaken to remove most of the NaOCl solution and placed under a running laminar flow bench to dry. They were further surface sterilised with ultraviolet germicidal lights behind ultraviolet resistant welding curtains on the laminar flow bench. After being cooked, but before the diets could set, 10 mL of diet was dispensed into each vial using a Jencons Scientific Perimatic GP II Peristaltic Pump Dispenser (Cambridge Scientific Products, Watertown, MA, USA). Each diet formulation had a replication of 100 vials. Fifteen vials prepared per diet formulation were used to determine development time to first pupation and the remaining 85 vials were for harvest at full pupation. The vials with their dispensed diet were left to cool for 1 h on a running laminar flow bench. Once cool, the diet surface in each vial was scarified using a sterilised dissecting needle (dipped in Denol (70%) and then in distilled water), breaking the skin of the diet surface, which allowed the enclosed larvae to enter the diet more easily. Two grams of sago (mixed with 0.0004 g of Dithane M45 as a fungicide) was poured over the surface of the diet in each vial. The sago helped absorb excess moisture on the surface of the diet and also served as a refuge for neonate larvae before they entered the diet.

#### 2.4.3. Inoculation of Neonate Larvae onto the Diet

*Eldana saccharina* eggs (oviposited on sheets of paper towelling) were collected from the SASRI-IRU adult emergence and oviposition room. The eggs were kept in a CLN 32 Laboratory Incubator (Pol-Eko Aparatura, Wodzislaw Slaski, Poland) for 7 days, at 24 °C, 72% RH and 0 L: 24-h D photoperiod, for them to hatch into neonate larvae. Two neonate larvae were carefully placed on top of the sago in each vial, using a fine paintbrush dipped in Denol (70%) and then distilled water. The vials were sealed with lids that provided proper ventilation through stainless steel fine mesh gauze to prevent neonates from escaping, labelled accordingly (i.e., diet formulation, inoculation date, quality assessment dates) and placed in plastic storage baskets. They were kept in a larval growth room (maintained at 26 ± 2 °C, 72 ± 5% RH and 0 L: 24-h D photoperiod). The plastic storage baskets were stacked on clean 5-tier metal racks. A maxim iButton DS1923 (configured using the Fairbridge Technologies ColdChain thermodynamics software) (Fairbridge Technologies, Sandton, South Africa) was placed in the larval growth room with the inoculated vials, hanging on a tag on a 5-tier metal rack, to monitor the temperature and humidity in the room at 15-min intervals.

#### 2.4.4. Physical Properties of Artificial Diets

The physical properties (pH, moisture content and water activity) of the different diets listed in Table 1 were tested once a week from the period of inoculation up until harvest. Diet contamination was also checked by visual observation. The most common contaminant is a green fungus (*Aspergillus* spp.) known to compromise diet quality and lead to increased larval mortality [25].

#### 2.4.5. Diet pH Testing

The diets’ pH was tested using a 2-point calibrated flat probe pH meter (Hanna Instruments, Woonsocket, RI, USA). The pH meter was first calibrated by inserting the probe into a neutral buffer solution (pH at 20 °C ± 0.1 °C of 7.00 ± 0.02). Once calibrated, the probe was rinsed with distilled water and inserted 3 cm into the diet of each vial. The pH was then recorded.

#### 2.4.6. Moisture Content Determination

A 50 g sample of each formulation was baked dry to constant mass at 65 °C for 5 days. It was weighed daily using a calibrated two decimal place balance until constant mass was obtained. The difference in initial and final mass was assumed to be moisture loss and expressed as a percentage of initial mass. Moisture lost by the diets over a 4-week period of larval growth was determined. Five empty vials per formulation (to be subtracted from the weights of the vials containing the diet, to give the weight of the diet sample only) were weighed, 10 mL of diet from each formulation was dispensed into each vial and 2 g of sago was poured over the surface of the diets. The vials containing the diet were then weighed to obtain initial weights. The diets were kept at 26 ± 2 °C, 72 ± 5% RH and 0 L: 24-h D photoperiod in a larval growth room. On a weekly basis the vials containing diet were re-weighed and recorded. Each week the weights obtained were subtracted from the previous weeks weights to determine mean moisture lost and the mean moisture lost over the 4-week testing period was subtracted from the initial moisture content (determined using the dry oven method).

#### 2.4.7. Water Activity Measurement

The water activity in the diets was measured using a Decagon Pawkit water activity meter (Lab Cell Ltd., Mansfield Park, Ireland). In disposable 8 mL Aqualab sample cups, 4 mL of diet per formulation were filled in each cup and the bottom of each cup was completely covered with the diet. Each prepared sample cup containing diet was placed onto a level surface. To take a measurement, the sensor cover of the Pawkit meter was flipped back and the Pawkit meter was placed onto the sample cup. The cup fitted over the sensors into a recess in the bottom of the Pawkit meter, making a vapour seal with the sensor. Once the Pawkit meter was properly positioned over the sample cup, readings of water activity within the diet samples were taken.

### 2.5. Quality Assessment

#### 2.5.1. Development Time to First Pupation

After 20 days in the larval growth room, life stages from the allocated 15 vials were manually and gently extracted from the diets in the vials using dissecting forceps, and divided into size categories (1st/2nd instar; 3rd/4th instar; 5th/6th instar; pre-pupal stage and pupal stage). These were placed into one of five 250 mL plastic jars with lids, according to its respective label. Each of the test diets was assessed in this way. The respective life stages were counted using a tally counter after the diet assessment was completed. Dead insects from each diet were counted as they were found. Once all life stages were counted, the results were recorded.

#### 2.5.2. Pupal Harvesting and Weighing

To determine pupal production, the remaining vials (85 vials per formulation) were harvested 27 days after inoculation. This period was previously determined to be peak pupal production time under similar laboratory conditions [7]. Details of the harvested vials were recorded (i.e., diet formulation; date of inoculation; date of harvest and number of vials harvested) and the insect life stages (i.e., larvae; pre-pupa; pupae; moths and dead insects) collected were then counted and recorded. Harvested pupae were placed singly into cells of multicell trays (*n* = 32 cells per tray) covered with ventilated plastic cling wrap film to prevent adult escape. Pupae were placed singly per cell to ensure that virgin males and virgin females were available for further experiments (i.e., oviposition testing; Section 2.6). Trays were labelled with diet formulation and harvested date, stacked on multicell tray metal racks and stored in the IRU adult emergence and oviposition room (maintained at 26 ± 2 °C, 72 ± 5% RH and 8-h L: 16-h D photo phase) on 5-tier metal racks. The trays were stacked on 5-tier metal racks to allow for sufficient ventilation and even distribution of temperature and humidity conditions.

Pupae carefully cut out of their cocoons were collected from harvested batches of pupae for each diet formulation. Using a light dissecting microscope, 30 male and 30 female pupae per formulation were identified, based on the different structures of their external genitalia [26]. The pupae of both sexes were weighed separately using a calibrated four decimal place balance (Microsep (Pty) Ltd., Johannesburg, South Africa).

#### 2.5.3. Adult Emergence and Sex Ratio

Adult emergence and sex ratio counts were completed from the pupae produced from the different diets. A day after harvest and on a daily basis thereafter (process took a maximum of 2 weeks), freshly emerged adult males and females were counted and recorded. Markings on the cells (M for males and F for females) were made to identify and separate the insects that had emerged. From the emergence, the ratio of males-to-females in each diet formulation was determined.

### 2.6. Oviposition Testing

#### 2.6.1. Male and Female Mating Frequency

To determine mating frequency of the males emerging from each diet, a freshly emerged virgin male and female moth pair was placed into a 500 mL paper drinking cup, containing a pleated cardboard oviposition substrate (50 × 10 mm when pleated 5 times) held together with a paper clip. A 10 mm dental cotton wick soaked with distilled water, for adults to drink from, was attached to the paper cup lids [9]. The lids were placed on the paper cups after the moth pair and oviposition substrate was placed in it, and labelled accordingly (i.e., diet formulation, harvested date, inoculated date). The paper cups were placed upright in plastic storage baskets and placed on 5-tier metal racks in the IRU adult emergence and oviposition room.

On a daily basis, the oviposition substrate and the female moth were removed and the water supply was replenished. A new oviposition substrate and a freshly emerged female were placed into the paper cup with the remaining male and left to mate overnight [9]. The oviposition substrate was not used for fecundity in this section but was placed in the cup to prevent the females from randomly laying eggs inside the cups. The female removed was killed by freezing and placed into a plastic resealable bag labelled with the male she was paired with and the date she was placed with the male. The female was then dissected under a light dissecting microscope to assess mating status by checking her *bursa copulatrix* for the presence of spermatophores [9]. The procedure was repeated with all the females presented to each male until the male died, to determine how many females the males from each diet had successfully copulated with in its lifetime. A total of 15 males per diet formulation were used.

To determine female mating frequency, the pairing of virgin female moths with virgin male moths, from each diet formulation, was prepared as described above. The oviposition substrate and the male moth was removed and the water supply was replenished daily. A new oviposition substrate and freshly emerged male moth were placed into the paper cup with the remaining female moth and left to mate overnight [9]. This procedure was repeated until the female died, after which she was dissected to assess mating frequency by counting the spermatophores within her *bursa copulatrix*, assuming the males she was paired with had transferred only one spermatophore to her *bursa copulatrix* on the night with her [9]. A total of 15 females per diet formulation were used to assess mean mating frequency.

#### 2.6.2. Female Fecundity and Fertility

To determine female fecundity, a single freshly emerged male and female moth pair (15 moth pairs per diet formulation) were placed into a 500 mL paper drinking cup, prepared as described in the mating frequency section. Oviposition substrates were replaced daily, and the water supply was replenished until the female moth died [9]. Each collected oviposition substrate was inserted into a plastic resealable bag, labelled accordingly (i.e., diet formulation, inoculated date, cup number, date of removal) and placed in an incubator, kept at 24 °C, 72% RH and 0 L: 24-h D photoperiod. To determine the mean fecundity per *E. saccharina* female, the total number of eggs oviposited on the oviposition substrates each day were counted under a light dissecting microscope for each female in the trial. These were summed for each female, and divided by the number of females in the trial, to get the mean fecundity of each female in the trial [9].

To determine mean fertility of the eggs oviposited per female, the total number of black head stage eggs [9] or neonate larvae emerging from the eggs oviposited by each female each day, were counted under a light dissecting microscope, 5 days after each oviposition substrate was removed from the cup, to allow the neonate larvae to develop in the eggs. Unfertilised eggs were also counted. The fertile and infertile eggs oviposited per day per female were summed, and the percentage of fertilised eggs were calculated per female. These were then summed for each female and divided by the number of females in the trial, to get the mean fertility of the eggs oviposited by each fertilised female in the trial [9].

### 2.7. Statistical Analyses

One-Way Analysis of Variance was performed using IBM SPSS Statistics version 22 (2013: IBM Corp., Costa Mesa, CA, USA), on the diet pH, moisture content and water activity data, to compare means between the different diet formulations listed in Table 1. ANOVA was also performed on the percentage survival and population age distribution (at the time to first pupation and at full pupal harvest), male and female pupal weight, moth emergence and sex ratio, male and female mating frequency and female fecundity and fertility data, to compare means between the different diet formulations listed in Table 1. Significant means were separated with a Tukey’s HSD post hoc test, at *p* < 0.05. Key assumptions of ANOVA were checked and met for homogeneity of variance (using the Levene’s test, at *p* > 0.05) and normality (using a Shapiro–Wilk test for normality) of data distributions. The SC and PAP diets described in Table 1 did not produce any results and thus were not included in the data analysis.

## 3. Results

### 3.1. Pysical Properties of Artificial Diets

The pH, moisture content and water activity did not vary between the diet formulations. pH: 4.79 ± 0.02 (*n* = 4) (*p* = 0.535), moisture content: 81.43 ± 1.07% (*n* = 4) (*p* = 0.807) and water activity: 0.92 ± 0.00 a_w_ (*n* = 4) (*p* = 0.102) over the 4-week period tested (Table 2). No biological contamination was observed in any of the diet formulations throughout the duration of the trial.

### 3.2. Insect Production during Quality Control and at Harvest

Survival from neonate to large instar/pupal stage did not vary between the diet formulations. Overall, survival was good, with a mean of 98 ± 1.14% (*n* = 15) (*p* = 0.243) developing in the time to first pupation trial, and a mean of 99 ± 0.33% (*n* = 85) (*p* = 0.370) surviving in the full pupal harvest trial (Table 3).

### 3.3. Population Age Distribution

After 20 days, the nutrients supplied by the IAAP3/4 diet and then the MS diet resulted in the fastest development of *E. saccharina* from neonates through to pupae (25% and 17% prepupae and pupae, respectively; Table 4). This is also evidenced by the lower percentage of larvae in the IAAP3/4 (1st/2nd Instar (0%), 3rd/4th instar (17%) and 5th/6th Instar (55%)) and the MS (1st/2nd Instar (2%), 3rd/4th instar (17%) and 5th/6th Instar (64%)) diets. Even though the control diet had a total of 20% pupae at the same time, the percentage of 5/6 instars (45%) and also the smaller instar (1st/2nd Instar and 3rd/4th instar = 25%) populations was greater than that in the IAAP3/4 and MS diets. The *E. saccharina* in the other diets tested showed significantly slower growth (Table 4).

After 27 days of development at the same constant temperatures, *E. saccharina* developed fastest in the IAAP3/4 and MS diets (93% and 92% pupae and moths, respectively). Even though 99% of the larvae had pupated in the control diet, no moths had emerged yet. *Eldana saccharina* in the other diets developed significantly slower at the same constant temperatures (Table 5).

### 3.4. Male and Female Pupal Weight

In all diet formulations, female pupal weight was close to double that of male pupal weight. Male pupal weight was significantly lower in the IAAP3/4 diet (weight: 0.1045 ± 0.01 g [*n* = 30]; *p* = 0.027) as compared with the control diet, but not significantly different from the pupae produced in the other diet formulations. Female pupal weight was significantly lowest in the MS (weight: 0.1864 ± 0.01 g [*n* = 30]; *p* = 0.021) and IAAP2 (weight: 0.1835 ± 0.01 g [*n* = 30]; *p* = 0.041) diets as compared with the control diet, but not significantly different from pupae produced in the other diet formulations (Figure 1).

### 3.5. Moth Emergence and Sex Ratio

Moth emergence from pupae harvested from all diet formulations was high (≥90%). The highest emergence was from pupae harvested from the MS diet (emergence: 98 ± 0.58% (*n* = 170); *p* < 0.001), followed by those from the IAAP3/4 (emergence: 97 ± 0.58% (*n* = 170); *p* < 0.001) and control (emergence: 96 ± 0.58% (*n* = 170); *p* < 0.001) diets. The sex ratio of adults emerging from all pupae harvested from the different diet formulations were all close to 1 (ratio: 1.0 ± 0.03 [*n* = 170]; *p* = 0.119) meaning that the sex ratios are not biased towards males nor females (Table 6).

### 3.6. Male and Female Mating Frequency

The mating frequency of the adult males and females from the different diet formulations varied, with males from the MS, IAAP3/4 and control diets mating with the least number of females (mating frequency: 4 ± 0.23 times (*n* = 15); *p* = 0.006, mating frequency: 4 ± 1.76 times (*n* = 15); *p* = 0.041 and mating frequency: 4 ± 0.21 times (*n* = 15); *p* = 0.012, respectively) and males from the IAAP5/6 diet mating with significantly more females (mating frequency: 6 ± 0.67 times (*n* = 15); *p* ≤ 0.041) (Table 7).

Females from the IAAP2, IAAP 3/4 and IAAP 5/6 diet formulations only mated once (mating frequency: 1 ± 0.00 times (*n* = 15)]; *p* = 0.021, mating frequency: 1 ± 0.00 times (*n* = 15); *p* = 0.021 and mating frequency: 1 ± 0.33 times (*n* = 15); *p* = 0.017, respectively), despite having the choice to mate with more males, while females from the MS diet formulation mated with significantly more males (mating frequency: 3 ± 0.19 times [*n* = 15]; *p* = 0.021) (Table 7).

### 3.7. Female Fecundity and Fertility

There were no significant differences observed in fecundity (871 ± 28.13 eggs (*n* = 15); *p* = 0.066) and fertility (94 ± 1.01% (*n* = 15); *p* = 0.783) of female moths produced from the diet formulations when mated with male moths from the same diet formulations (Table 8).

## 4. Discussion

There has been considerable progress in the development of artificial diets for *E. saccharina* since it became a recurring pest in South African sugarcane in the 1980s [10]. Each diet showed improved production and cost benefits over the previous ones. However, none had been based on the carcass milling principles used in animal science to develop feed for livestock and poultry [14]. This approach is used to formulate animal diets based on the animals’ actual nutrient requirements, and the actual nutrients that their natural foods provide [14]. Recent research demonstrated that this approach could be used to develop high quality and cost-effective diets for insects [13,15,16]. The study reported on in this paper complemented the work of Woods et al. [13] and used the CMT to develop a cost-effective larval diet that did not compromise the quality of *E. saccharina*. The current diet used to rear *E. saccharina* [10] was developed from a previously published diet for *O. nubilalis* [22] and proved to be more efficient and cost-effective than the previous diet developed by Walton and Conlong [9], but it did not take into account the actual nutrient requirements needed for good development of *E. saccharina*.

### 4.1. Are the Formulated Diets, Derived from the CMT, Similar or Better at Rearing Eldana saccharina than the Control Diet Used in the Current Study?

In terms of the artificial diets developed using the CMT, their physical properties all proved as good as the previous diets, as reflected by the actual growth of *E. saccharina*.

The pH of the diets formulated according to the CMT and the control diet was found to be between 4.76 and 4.85, with an average of 4.79. Similar published results were recorded by Ngomane et al. [10] and Woods et al. [13] who found an average pH of 4.9 and 4.7, respectively, in the diets they formulated. In addition, these reflected the pH of *E. saccharina’s* natural host plant, sugarcane (4.5–5.5; [27]). This is in accordance with Cohen’s [3] generalisation that insects require a slightly acidic pH range in their diet. In addition, the moisture content of the CMT and control diets was retained between 79.86 and 83.34% which was slightly higher than the moisture content of sugarcane (68–74%; [28]). According to Cohen [3], most plant-feeding insects require high water content (between 70 to 90%) in their diets to sustain their life processes and without an adequate amount of water all life processes fail. In natural foods, water is retained within cell walls and bound to cellular constituents, and in artificial diets a nutritionally inert substance such as agar or carrageenan gel is required to bind water. In the current study, *E. saccharina* larvae did not survive on the SC and PAP natural host plant diets developed using the CMT and therefore were not reported on. In a similar study conducted by Woods et al. [13] the SC and PAP diets lacked the ability to bind water which supported evaporation, but when the moisture content of the diet became too low, the larvae were unable to gain access to nutrients, resulting in poor development and increased mortality of *E. saccharina* feeding on the diet.

Water activity of the CMT-derived and control diets was found to be between 0.90 and 0.93 a_w_ and no biological contamination was observed in all diet formulations. According to Rockland and Nishi [29], microorganisms (i.e., bacteria, mould, yeast etc.) require free water for growth and every microorganism has a minimum water activity below which it will not grow. At a water activity lower than 0.95, micro-organisms are generally inhibited because there is not enough free water available to support pathogen growth [29] and this was evident in the diets formulated in this study, as no contamination was recorded.

In the light of this, the physical properties (pH, moisture content and water activity) of the diets formulated using the CMT were suitable for the growth of *E. saccharina*. This is further supported by the improved biological parameters of *E. saccharina* reared on these diets.

### 4.2. Effect of the Different Diet Formulations, Based on the CMT, on Biological Parameters of Eldana saccharina

#### 4.2.1. Insect Development

The results obtained from the CMT diets (except for the SC and PAP formulations, as already explained) have revealed that *E. saccharina* can develop from neonate larvae to the adult stage, with a survival rate of 97–100% in all diet formulations, indicating that all diet formulations provided the required nutrients for larval development. Feeding the larvae with the MS and IAAP3/4 diets shortened the time to first pupation, from 27 [10] to 20 days, compared with larvae fed on the remaining CMT diets and the control diet. The duration of the larval development time was reduced using these diets, compared to the larval development times, at the same temperatures, recorded by Gillespie [18] and Way [30] on the old sugarcane-based diet of Graham and Conlong [19]: 35 and 43 days, respectively. Differences in growth rates between diet formulations may have resulted from differences in the quality and quantity of available nutrients which also depend on the quality and proportion of ingredients in each formulation [31]. For most laboratory rearing, larval diets are composed of a mix of ingredients (as demonstrated by the diets developed in this study) and these ingredients are the costliest components of the process [31]. *Eldana saccharina* mass-rearing requires cost-effective production and therefore, mass-production would become more affordable and suitable if larval feeding can be minimised, while ensuring that the quality of insects produced remains high. A diet that can shorten the period of larval development will help reduce the costs of the diet, labour and accelerate the production of adults [32]. This is important when considering mass rearing *E. saccharina* adults for a SIT programme. Rearing *E. saccharina* larvae on the MS and IAAP3/4 diets will help provide more generations per year, thus giving more insects that can be sterilised and released into the field [32].

#### 4.2.2. Pupal Weight

Within the dietary formulations, female pupae were found to be heavier than male pupae, supporting the findings of Walton and Conlong [9], Ngomane et al. [10] and Woods et al. [13]. The CMT diets did not significantly influence male and female pupal weights as compared with the control diet (previously observed by Woods et al. [13]) and that published by Ngomane et al. [10]. However, male and female pupae produced from these diets were found to be heavier than those produced by the diet developed by Walton and Conlong [9], proving that the diets developed in this study were of superior quality. The highest pupal weight was recorded with females from the IAAP3/4 diet and males from the MS diet. Female pupal weight is directly proportional to their fecundity and therefore, heavier female pupae will result in bigger adults, providing more eggs oviposited [25]. Large males on the other hand produce large spermatophores. The males prefer mating with large, young and virgin females, and the females prefer mating with large and mid-aged males. Mating of larger parents produces larger offspring [33]. Thus, the production of heavier *E. saccharina* females will be beneficial for a SIT programme, in that fewer females mated with the larger males can be used to produce high numbers of good quality insects. The production of high quality and heavier partially sterile males would also increase their competitive ability with wild males [34], and thus increase their impact in SIT programmes.

#### 4.2.3. Moth Emergence and Sex Ratio

The CMT diets proved to be of highest quality for rearing *E. saccharina*, resulting in adult emergence rates of 90 to 98%. The highest emergence came from pupae harvested from the MS diet. This shows an increase of at least 8% in adult emergence compared to that obtained with the diet used by Walton [35] that resulted in an adult emergence of 89.2%. Since large scale SIT programmes depend on irradiation of mature pupae or freshly emerged adults, it is important to provide an efficient artificial diet for *E. saccharina* larvae that produces enough pupae or moths to supply the overflooding ratios needed for SIT [10].

When mass-rearing for releases in control programmes and producing insects for experimental purposes, a sex ratio that optimises production of both male offspring (e.g., for SIT) and female offspring (e.g., for mass rearing purposes) is required in the breeding population [36]. In the current study, an even male-to-female ratio of 1:1 of emerged *E. saccharina* moths was observed for all diet formulations, which according to Sampson and Kumar [37] is slightly different from that found in field populations.

#### 4.2.4. Mating Frequency

Efficacy of the SIT relies upon released sterile male insects efficiently transferring their sperm, which carries dominant lethal mutations, to wild females [38]. Released irradiated females do not play an important role in population suppression when deployed in a programme with an Inherited Sterility (IS) component [39]. Thus, success or failure of the technique largely depends on the quality of sterile males and their ability to locate and mate with wild females [38]. Understanding mating behaviour of species targeted for the SIT, and more specifically their mating systems and how mass-rearing and irradiation impacts them, are important steps that can lead to improvements in sterile male performance. This could possibly reduce sterile to wild male overflooding ratios routinely applied to compensate for the lower effectiveness of mass-produced sterile insects [38]. Should the released males be as competitive as wild males, rearing costs could be significantly reduced [35].

Walton [35] reported that a radiation dose of 200 Gy does not compromise the performance of *E. saccharina* males in an IS programme. Mudavanhu et al. [38] further investigated mating competitiveness and compatibility of non-irradiated and irradiated *E. saccharina* moths under laboratory and semi-field conditions. They found no mating barriers between released irradiated and non-irradiated populations in their studies. The released irradiated males were able to successfully compete with non-irradiated males for wild females, and they also observed that wild females did not discriminate against irradiated or non-irradiated moths which indicated no negative effects due to laboratory rearing or radiation.

In the current study, each *E. saccharina* male reared on the CMT and control diets showed the ability to mate more than once in the laboratory when presented with a freshly emerged virgin female on consecutive nights during their life span. The highest mating frequency was recorded with males reared on the IAAP5/6 diet which mated with an average of six different females. Similar observations were made by Walton and Conlong [9] who reported males mating with an average of 3.3 (maximum of 6) females. The ability of *E. saccharina* males to mate multiple times has important implications for calculating the required release rates of sterilised male moths in order to obtain sufficient sterile to wild male overflooding ratios needed for SIT [35].

When given the opportunity to mate with more than one male, females from the CMT diets mated with an average of 1.5 males. The highest mating frequency was recorded with females reared on the MS diet which mated with an average of three different males. Again, these results were in line with those of Walton and Conlong [9] who reported females mating with an average of 1.5 (maximum of three) males. The ability of *E. saccharina* females to mate more than once in the laboratory is a good indication for mass-rearing purposes, in that females reared on the MS diets will be able to mate multiple times with fertile males, producing more offspring. Although females from the MS diets mated multiple times, the majority of females reported in this study mated only once. It is likely that females in the wild will mate once [26], and if sterile males successfully mate with wild females, this will reduce the chance of wild females accepting mating with wild males [35].

#### 4.2.5. Fecundity and Fertility

Mean fecundity and fertility of *E. saccharina* females reared on the CMT and control diets were very similar to that of Dick [40] and Betbeder–Matibet et al. [41] who reported a mean of 750 eggs per female that was 97% fertile. Fecundity and fertility of *E. saccharina* females obtained in this study were significantly higher than that obtained by Gillespie [18] who reported an average of 318 eggs per female, reared on the diet of Graham and Conlong [19], and Walton and Conlong [9] who reported an average of 518 eggs per female that were 63% fertile, from adults reared on their diet. Ngomane et al. [10] showed that the Walton and Conlong [9] diet was inferior in terms of quality and insect production as compared with the *O. nubilalis*-based diet formulation. On the strength of this, it is clear that the experimental diets developed in the current study were of a superior quality as compared with the diet produced by Walton and Conlong [9], resulting in better quality insects being reared from them. The most significant effect that increased fecundity and fertility of *E. saccharina* females would have on an SIT programme is the reduction of the costs of mass-rearing, as fewer females that produce many fertile eggs will be required to produce more insects that need to be sterilised and released into the field.

The CMT proved to be effective at developing two high quality diets for mass-production of *E. saccharina* that did not compromise insect growth, development and reproduction. The development period of larvae feeding on the MS and IAAP3/4 diets was significantly reduced as compared with the larval development period of the control in this study, and that reported by Ngomane et al. [10]. When comparing the rest of the biological parameters presented above, the MS and IAAP3/4 diets performed similarly to that of *E. saccharina* reared on the control diet, and the published diet of Ngomane et al. [10]. This implies that larval rearing on the MS and/or IAAP3/4 diets at the SASRI-IRU will accelerate the production of adults throughout the year, leading to a greater, constant supply of insects for use in a SIT programme against *E. saccharina*.

## 5. Conclusions

Success of the SIT, as part of an AW-IPM programme against economically important insect pests, largely depends on the ability to mass-produce high quality insect species and establish their colonies under controlled laboratory conditions [1]. A rearing system has been developed for mass-production of *E. saccharina* at SASRI. Although mass-production of this insect has been set in place, improvements of rearing procedures and developing more cost-effective diets for mass-rearing purposes and improving the quality of the artificial diets for better production of good quality insects is necessary to optimise *E. saccharina* production in this facility. This will enable a constant and large supply of good quality insects needed for effective field releases in a SIT programme.

In the current study, experiments to achieve the above objectives were conducted to evaluate the growth and performance of *E. saccharina* on diets developed using the CMT. The results demonstrated that *E. saccharina* performed best when reared on the minimum specification (MS) diet and the diet resembling the ideal amino acid profile of the 3rd–4th instar larvae (IAAP3/4). Feeding *E. saccharina* larvae with the MS and IAAP3/4 diets shortened their development period from 27 days to 20 days as compared with the control diet, whilst still maintaining good pupal mass and female adult fecundity, and not impacting adult sex ratio and mating ability. Based on these findings, it is clear that the MS and IAAP3/4 diets most closely suited *E. saccharina* growth and development. Even though both the MS and IAAP3/4 diets provided faster development at day 20, the MS diet is the preferred choice to replace the current diet used to rear *E. saccharina* at SASRI, as this diet resulted in the highest percentage of moths (9%) on day 27, and a higher adult emergence (98%) than the IAAP3/4 diet (7% and 97% emergence, respectively), proving that larval development is faster on this diet than the IAAP3/4 diet.

## Figures and Tables

**Figure 1 insects-13-00316-f001:**
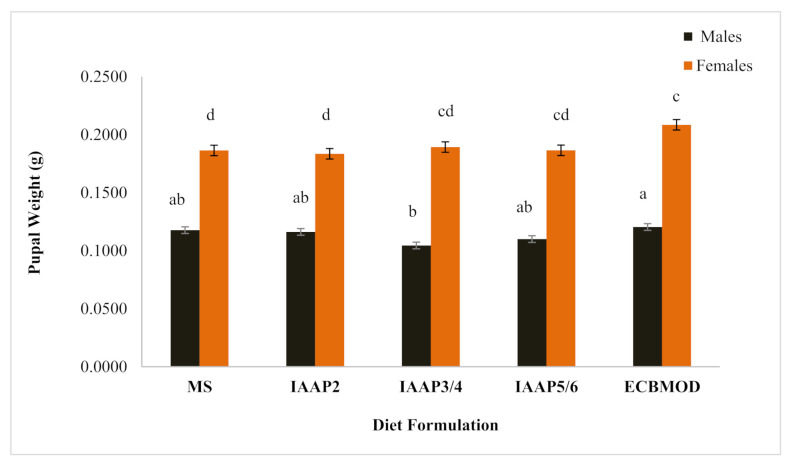
Mean (±SE) *Eldana saccharina* male and female pupal weights (*n* = 30) at harvest (day 27) from the minimum specifications (MS), ideal amino acid profile of second instar *E. saccharina* larvae (IAAP2), ideal amino acid profile of third and fourth instar larvae (IAAP3/4), ideal amino acid profile of fifth and sixth instar larvae (IAAP5/6) and control (ECBMOD) diets. Different lower-case letters above the graph histogram bars indicate that mean differences are significant at the 0.05 level.

**Table 1 insects-13-00316-t001:** Ingredients and weights (Wt) required for preparing the formulated *Eldana saccharina* and Control diets (MS = minimum specifications diet; IAAP2 = ideal amino acid profile of second instar *E. saccharina* larvae based diet; IAAP3/4 = ideal amino acid profile of third and fourth instar larvae based diet; IAAP5/6 = ideal amino acid profile of fifth and sixth instar based diet; SC = diet based on the analysis of sugarcane (host plant); PAP = diet based on the analysis of *Cyperus papyrus* (host plant).

Formulated Diets vs. Control Diet								
Ingredients	Diet Name	MS	IAAP2	IAAP3/4	IAAP5/6	SC	PAP	CONTROL
	Unit	Wt.	Wt.	Wt.	Wt.	Wt.	Wt.	Wt.
Carrageenan gel	g	15.00	15.00	15.00	15.00	15.00	15.00	
Agar powder	g							4.60
Lucerne meal	g	250.00	250.00	250.00	250.00	50.00	50.00	
Rabbit meal	g							226.40
Wheat bran	g	55.80	53.20	6.00	6.00		40.75	56.60
Yeast extract	g							3.40
Ground chickpea	g	53.00	54.60	85.20	85.20	84.75	81.30	56.60
Full cream milk powder	g	7.60	7.60	7.60	7.60	7.60	24.45	22.60
Whole egg powder	g	28.40	28.40	28.40	28.40	28.40	28.40	28.20
Sucrose	g	66.20	66.40	68.60	68.80	70.25	133.00	64.60
Sodium chloride	g					4.72	63.00	0.60
Nipagin	g	6.40	6.40	6.40	6.40	6.40	6.40	6.40
Sodium propionate	g	10.40	10.40	10.40	10.40	10.40	10.40	10.40
Oxytetracycline	g	2.00	2.00	2.00	2.00	2.00	2.00	2.00
Ascorbic acid	g	6.40	6.40	6.40	6.40	6.40	6.40	6.40
Acetic acid	mL	8.00	8.00	8.60	8.00	8.60	8.60	8.00
Citric acid	g	2.60	2.60	2.60	2.60	2.60	2.60	2.60
Tri-sodium citrate	g	2.60	2.60	2.60	2.60	2.60	2.60	2.60
L-Lysine HCL	g						25.00	
Vit + min premix	g	0.80	0.80	0.80	0.80	0.80	0.80	
Total		515.20	514.40	500.60	500.20	300.52	500.70	502.00
Water for agar	mL							500.00
Water balance	mL	1500.00	1500.00	1500.00	1500.00	1500.00	1500.00	1000.00
Total diet volume	mL	2015.20	2014.40	2000.60	2000.20	1800.52	2000.70	2002.00

**Table 2 insects-13-00316-t002:** Mean (±SE) pH, moisture content and water activity of the MS, IAAP2, IAAP3/4, IAAP5/6 and control diets from the period of inoculation to harvest (4-week testing period).

Diet Formulation	Diet pH	Moisture Content (%)	Water Activity (a_w_)
MS	4.78 ± 0.04	81.06 ± 3.30	0.90 ± 0.01
IAAP2	4.76 ± 0.04	79.86 ± 2.59	0.92 ± 0.01
IAAP3/4	4.78 ± 0.06	83.34 ± 1.56	0.93 ± 0.01
IAAP5/6	4.80 ± 0.01	82.92 ± 2.57	0.93 ± 0.01
CONTROL	4.85 ± 0.03	80.00 ± 2.51	0.93 ± 0.01

**Table 3 insects-13-00316-t003:** Mean (±SE) survival of *Eldana saccharina* life stages reared on the MS, IAAP2, IAAP3/4, IAAP5/6 and control diets from inoculation of neonates to first pupation (*n* = 15) on day 20 and full pupal production at harvest (*n* = 85) (day 27).

Diet Formulation	% Survival at First Pupation (Day 20)	% Survival at Full Pupal Development (Day 27)
MS	100 ± 0.00	99 ± 0.60
IAAP2	100 ± 0.00	99 ± 0.83
IAAP3/4	97 ± 3.33	98 ± 1.31
IAAP5/6	100 ± 0.00	100 ± 0.00
CONTROL	93 ± 4.54	100 ± 0.00

**Table 4 insects-13-00316-t004:** Mean (±SE) distribution of *Eldana saccharina* life stages (percentage of the 1st/2nd instar larvae; 3rd/4th instar larvae; 5th/6th instar larvae; pre-pupae; pupae and mortality) [*n* = 30] surviving on the MS, IAAP2, IAAP3/4, IAAP5/6 and control diets, 20 days after inoculation. Means within columns with different lower-case letters indicate significant differences (*p* < 0.05).

Life Stage Distribution (%)						
Diet Formulation	1st/2nd Instar	3rd/4th Instar	5th/6th Instar	Pre-Pupae	Pupae	Mortality
MS	2 ± 0.58 c	17 ± 0.58 c	64 ± 0.58 a	5 ± 0.58 b	12 ± 0.58 b	0 ± 0.00 c
IAAP2	4 ± 0.58 c	39 ± 0.58 a	48 ± 0.58 c	0 ± 0.00 c	9 ± 0.58 c	0 ± 0.00 c
IAAP3/4	0 ± 0.00 c	17 ± 0.58 c	55 ± 0.58 b	5 ± 0.58 b	20 ± 0.58 a	3 ± 0.58 b
IAAP5/6	10 ± 0.58 b	25 ± 0.58 b	50 ± 0.58 c	5 ± 0.58 b	10 ± 0.33 bc	0 ± 0.00 c
CONTROL	15 ± 0.58 a	10 ± 0.58 d	45 ± 0.58 d	10 ± 0.58 a	10 ± 0.58 bc	10 ± 0.58 a

**Table 5 insects-13-00316-t005:** Mean (±SE) distribution of *Eldana saccharina* life stages (percentage of larvae, pre-pupae, pupae, moths and mortality) [*n* = 170] recorded at the time of full pupal harvest (day 27) on the MS, IAAP2, IAAP3/4, IAAP5/6 and control diets. Means within columns with different lower-case letters indicate significant differences (*p* < 0.05).

Life Stage Distribution (%)					
Diet Formulation	Larvae	Pre-Pupae	Pupae	Moths	Mortality
MS	2 ± 0.58 c	5 ± 0.58 b	83 ± 0.58 d	9 ± 0.58 a	1 ± 0.58 a
IAAP2	5 ± 0.58 b	8 ± 0.58 a	80 ± 0.58 e	6 ± 0.58 b	1 ± 0.58 a
IAAP3/4	2 ± 0.58 c	3 ± 0.58 bc	86 ± 0.58 c	7 ± 0.58 ab	2 ± 0.58 a
IAAP5/6	10 ± 0.58 a	1 ± 0.58 c	89 ± 0.58 b	0 ± 0.00 c	0 ± 0.00 a
CONTROL	0 ± 0.00 c	1 ± 0.58 c	99 ± 0.58 a	0 ± 0.00 c	0 ± 0.00 a

**Table 6 insects-13-00316-t006:** Mean (±SE) emergence and sex ratio (male: female) of *Eldana saccharina* [*n* = 130] from pupae harvested from the MS, IAAP2, IAAP3/4, IAAP5/6 and control diets. Means within columns with different lower-case letters indicate significant differences (*p* < 0.05).

Diet Formulation	Moth Emergence (%)	Sex Ratio (M:F)
MS	98 ± 0.58 a	0.9 ± 0.06
IAAP2	91 ± 0.58 b	1.1 ± 0.06
IAAP3/4	97 ± 0.58 a	1.1 ± 0.06
IAAP5/6	90 ± 0.58 b	1.0 ± 0.06
CONTROL	96 ± 0.58 a	1.1 ± 0.06

**Table 7 insects-13-00316-t007:** Mean (±SE) mating frequency of male and female *Eldana saccharina* (*n* = 15) from the MS, IAAP2, IAAP3/4, IAAP5/6 and control diets. Means within columns with different lower-case letters indicate significant differences (*p* < 0.05).

Diet Formulation	Male Mating Frequency	Female Mating Frequency
MS	4 ± 0.23 b	3 ± 0.19 a
IAAP2	5 ± 0.00 ab	1 ± 0.00 b
IAAP3/4	4 ± 1.76 b	1 ± 0.00 b
IAAP5/6	6 ± 0.67 a	1 ± 0.33 b
CONTROL	4 ± 0.21 b	2 ± 0.21 ab

**Table 8 insects-13-00316-t008:** Mean (±SE) fecundity of female *Eldana saccharina* (*n* = 15) and their egg fertility after mating with moths from the same formulations, from the MS, IAAP2, IAAP3/4, IAAP5/6 and control diets.

Diet Formulation	Fecundity (*n*)	Fertility (%)
MS	917 ± 40.06	94 ± 1.65
IAAP2	764 ± 73.14	96 ± 1.17
IAAP3/4	991 ± 105.76	96 ± 2.83
IAAP5/6	963 ± 70.07	94 ± 5.05
CONTROL	789 ± 48.66	94 ± 1.77

## Data Availability

Data supporting the reported results are the property of SASRI, and can be requested from the Programme Manager listed under the Crop Protection Program of SASRI (https://sasri.org.za).

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
