# Peer review of "Formulation of Artificial Diets for Mass-Rearing Eldana saccharina Walker (Lepidoptera: Pyralidae) Using the Carcass Milling Technique"

_insects, 2022, doi:10.3390/insects13040316_

Round 1
Reviewer 1 Report
Nomalizo C. Ngomane, Elsje Pieterse, Michael J. Woods and Des E. Conlong
Formulation of artificial diets for mass-rearing Eldana saccharina Walker (Lepidoptera: Pyralidae) using the Carcass Milling Technique
This is an excellent paper that reports a successful approach to optimization of an insect diet by using analytical techniques and empirically-based adjustments. The paper provides very useful information based on meticulously-conducted techniques. The paper is definitely of such quality that it should be published, but I have some comments about terminology that I think needs to be revised, and there are a few previous references that should be included.
The mention of energy (Lines 55, 56, 69, and 73) is misleading because this is not addressed or analyzed in elsewhere in this paper. While energy metabolism is of key importance in insects and other organisms, it does not seem to be (at least as demonstrated in this paper) the basis of success or failings of the diets being developed and compared.
The authors mention COST as a parameter that they fed into the system to help optimize the economic and the diet quality features (Line 195). However, though they mention cost reduction many times throughout the manuscript, I have found no conclusion regarding the cost of the various diets that they tested and no relationship shown as to how much the different diets were possibly saving. I am left uncertain about how much something like the milk product, chickpea, or wheat bran (for example) influenced the cost vs. nutritional value of the diets.
Specific comments: the use of the expression “carcass milling diets” is problematic for me. First, I find it confusing because it is described by the authors as being based on whole carcass analysis and analysis of the target species’ natural food. It is not, in my opinion, primarily a milling technique (which implies size reduction procedures such as crushing, grinding, cutting, etc.) The word “milling” describes only a part of the analytical process, but indeed it is the analytical process that the authors describe as the way they arrived at the composition of diets.
I developed a kind of key to the terms to help me remember which diet is which. However, I am suggesting that the authors use more consistency and clarity in explaining the terms and in using them throughout the text.
Key to Diet Terms:
CMT = carcass milling technique
MS = Minimum specifications diet (“minimum ingredient specifications of published Eldana saccharina diets” Lines 28-29
IAAP2 = ideal amino acid composition and profile (Line 30) for 2nd instars
IAAP3/4 = ideal amino acid composition and profile (Line 30) for 3rd instars
IAAP5/6 = ideal amino acid composition and profile for 5th and 6th instars
ECBMOD = Control diet that had been used in previous rearing operations.
Line 216: These terms can be confusing to the reader. Why not simplify this a little by using the term Control Diet instead of “ECBMOD Diet”?
As a further general comment about this paper, I see two major concepts emerging, and while these concepts are very important and potentially useful to the insect rearing community, they need to be treated much more consistently and clearly. 1) The authors have used an analytical technique that goes back into insect literature well-before the citations mentioned by the authors (such as Rock, G. C. and K. W. King. 1966. Amino acid composition of the red-banded leaf roller, Argyrotaenia velutinana (Lepidoptera: Tortricidae) during development. Ann. Entomol. Soc. Am. 59: 273-275. Cohen, A. C. 1992. Using a systematic approach to develop artificial diets for predators. In Advances in Insect Rearing for Research and Pest Management (T. E. Anderson and N. C. Leppla, Eds.) pp. 77-91. Westview Press. Boulder, CO, as well as the discussion of this topic in Cohen 2015). 2) The authors also use the feed optimization model from WinFeed, and the potential value and creativity of this approach seems to me to become lost in the lumping of the processes into the label “carcass milling technique,” (which the authors sometimes refer to as a “cadaver” technique.) The analytical approaches to arriving at a target diet composition includes whole carcass (or simply carcass) analysis and food analysis (or natural diet analysis), AND the authors are coupling these analytical techniques with the optimization approach through use of the WinFeed software. I salute the authors for their creative/pioneering efforts to apply the two approaches (analysis and optimization statistics), and I respect the difficulties of using these techniques in the complex arena of how the insects respond to these diet formulation approaches. I also see that there are efforts to compare the incipient diets to existing diet technology, but there are some complexities that come from this that cause me to get lost in excess (and sometimes non-precise) terminology (such as the actual meaning of the “MS Diet” and the “ECBMOD Diet.”
The term, Ideal Amino Acids needs clarification. They appear to me to be more of a “target” amino acids, rather than “ideal.” The authors call the analysis profiles “Ideal” which is misleading. The word “ideal” suggests something that is proven to be the best, so the implication is that whatever the composition is (in terms of free and residue amino acids, for example) would be the BEST and most definitive formulation. A more neutral term such as amino acid target profile would better allow for the concept that the authors are searching for an optimal profile in terms of kinds and proportions of the various amino acids.
The optimization process: The authors used WinFeed software as an optimization process. This is a creative and potential useful approach that deserves more explanation. In Line 183, the reference “EFG, 2005” is not clear. I could not find it elsewhere in my search of the paper, so I think that other readers would be confused by this or would have problems duplicating the authors’ work.
Lines 135-145 and 147 “natural diets”
Proximate analyses (Lines 160-177): The lipid extraction procedure should be further characterized here, rather than referring to Reference [13]. Was it a Soxhlet extraction, a goldfish extraction, a phase-extraction where the chloroform at the bottom was aspirated and weighed, or what?
Line 183: It is not clear to me
Lines 239, 253 and Table 1: the concentration of acetic acid should be specified as glacial acetic acid or ?- Normal acetic acid.
Host plant analysis: Lines 135-150: The analysis of the host plants (sugar cane stalks) is somewhat problematic to me. As Cohen 2015 discusses, insects feed on portions of their food source, rarely on the entire food material. For a stem borer, I would think that the E. saccharina would feed on selected portions of the stems but not all of the stems, for example the cuticle and other outer parts of the sugarcane stalk. If this is the case, depending on how much of the stalk the larvae eat, the analysis is for extraneous portions of the host material. This point should be clarified and discussed further.
The word “cadaver” is not appropriate and should be replaced with “carcass.” (Lines 147, 162, 171).
Some comments about Table 1: It is not clear to me what was the rationale was for addition or removal several of the ingredients. For example, rabbit meal was eliminated in all but the ECBMOD diet as was yeast extract. Sodium chloride was added only to the SC, PAP, and ECBMOD diet (in the case of the PAP diet was, according to my calculation, about 12-13% of the diet’s dry mass). Why was lysine added to the PAP diet and not the others, and why lysine rather than the other amino acids, especially the essential amino acids?
Specific comments by lines:
Lines 122-126: The authors state that the heating in boiling water (known in the literature as “coddling”) was timed to stop metabolic activity but not to denature the proteins. The proteins DO become denatured at this temperature (i.e. their tertiary structure is permanently changed). I would suggest that the explanation of the timing be presented in terms of using the temperature and timing to prevent metabolic activity but not so much heating that the amino acid structures would, for the most part, be preserved. Indeed, when the authors subjected the insects to the protein hydrolysis conditions in 6 N HCl at 110oC for 24 hours (Line 155) they were subjecting the proteins to harsher conditions that the coddling process. In fact under the hydrolysis conditions, all tryptophan is destroyed along with much threonine, cysteine and methionine).
Line 223: lysine is an essential amino acid, but it is not a protein
Line 539: Here, the authors use the reference to “the diet of Ngomane et et.” but they do not refer to the diets’ names that are used throughout this paper (MS, IAAP2, etc.), so the reader must make the translation. It would be very helpful if the author were to use consistent terms throughout the paper once each diet name and type is described.
Lines 565-569: The generalization about water activity below 0.95 inhibiting micro-organisms’ growth is not accurate for may micro-organisms such as many yeasts and other fungi (which are often the MAJOR contaminants of insect diets). This point is made by Cohen 2015. An important point about the discussion in Lines 553-572 is that it does not adequately discuss the differences (or lack thereof) in performance on the various diets. This section seems to overly relegate the success or failures of the various diets to the “carcass milling technique.” However, in my reading, I cannot see how this technique relates to pH, moisture content, and water activity. As part of the technique, the authors describe drying the diets before the analytical procedures are conducted, which would obscure moisture and water activity factors (and possibly pH factors).
Lines 716-726: The authors point out that the best diet was the MS formulation, but it is not clear to me that this diet was developed with the optimization techniques or as a compilation of the diets from existing publications. Also, in Lines 713-716, the authors say that these studies were intended “to evaluate nutritional requirements and formulate better diets….” However, I do not find discussions or conclusions about the nutrient requirements in this paper. I think that the paper makes an excellent and important contribution to methods of optimization of complex diets, but it somewhat blunts this point by extending the interpretation to include nutrient requirements as a goal. Later in this paragraph, the authors say that the MS and IAAP3/4 diets most closely represented the nutrient requirements of E. saccharina, but this simplification of the outcome does not take into account such factors as differences in the palatability of the various diets nor of the bioavailability of the diets. It also misses the metabolic factors that may be making the diets perform differently of E. saccharina. In my trying to interpret the possible differences between the diets, I consulted Table 1 and tried to visualize which factors make the various diets different from one-another. The major differences that I see in this table are that 1) lucerne meal is present in all the diets EXCEPT for the ECBMOD diet; 2) the wheat bran quantities differ substantially; 3) the chickpea quantities are somewhat different; 4) the full cream milk powder amounts differ ONLY between the 1st 4 diets in the table and the last two diets. It would be helpful to the reader to get some guidance from the authors in interpreting how these differences might affect the major qualities of the diets that would make them more or less successful (from Cohen’s 2015 discussion: palatability, nutritional value, bioavailability, and possibly stability).
Author Response
Please see the attachment. Responses are marked as track changes.

Reviewer 2 Report
Title: Formulation of artificial diets for mass-rearing Eldana saccharina Walker (Lepidoptera: Pyralidae) using the Carcass Milling Technique.
Authors: Nomalizo C Ngomane, Elsje Pieterse, Michael J Woods and Des E Conlong
Summary:
Overall, this is a well designed and carefully executed study that has been well explained to the reader. There was obviously a lot of work undertaken here and the level of detail in the material and methods section highlights this fact. Studies such as this one, are important in ensuring operational SIT programs become and remain cost-effective whilst maintaining a high quality of insect. This study presents strong evidence from a series of well-thought-out experiments that offer an innovative way to enhance the diet of E. saccharina.
General Comments:
- Whilst not necessarily a criticism, you only present one graph in the results section and the rest in tabular format. I would consider adding another graph or two since you have a lot of information to get across within your study.
Specific Comments:
Introduction
L31: “The” control diet.
L65-67: This sentence could read better, it needs “the” to be inserted at some point. Perhaps remove “to help” and replace with “in order to improve the formulation of insect diets”.
L68-71: Again, I think the order of this sentence is wrong – it may be better to introduce the carcass milling technique at the start of the paragraph, which is a method of comparative slaughter, which has been used previously to…..now insect diets etc…
L70: If you use the abbreviation CMT in the abstract, then after the first full use of it within the main text, it should be abbreviated to CMT throughout the remainder of this manuscript.
L80-82: Remove “for the purpose they were developed”, this is stating the obvious. Maybe say something like “all diets” at the start of the sentence.
L87: Replace “again” with “further” or something similar, “improved” can be switched out for “refined/ enhanced”.
L88-89: “developed” used twice in the one sentence – try exchanging one of them for a similar word.
L90: Remove the second mention of “formulated” – it reads better without it.
L93 & 98: Use abbreviation CMT.
Materials & Methods
L109: You must state the abbreviation of relative humidity (RH) at the first use of it if you plan to use it later (L110). Be consistent throughout the text and check all abbreviations and their use. At this point I will stop commenting on the same thing.
L111: Please remove the word “photophase” and replace with “photoperiod”. Photophase only refers to daylight hours, and you mention both light and dark hours so photoperiod is more accurate. Please replace throughout remainder of the manuscript.
L113: You state the larvae were the life stage “subjected” to the diet – I’d choose another word, it seems like it’s being forced on them.
L122-125: Rearrange to say something like “the larvae were transferred to boiling water for 1 minute”. It just reads better.
L147: Larval cadaver? Cadaver is normally when referring to a deceased human, not animal, especially an insect.
L162: Remove cadaver again and throughout the remainder of the manuscript.
L216: Table 1 – although it is ok to say “amounts” you use Wt for weight underneath in the table so maybe change amounts to weights in the table title.
L258-261: I’d suggest splitting this into two sentences.
L264: You end the sentence with “100 vials” and start the next sentence with “fifteen vials”. Be consistent throughout with your use of numbers and text.
L281: Micron size of mesh?
L354: “weighed separately”.
Results
L442: Table 3 – has (Day and then nothing. What is this supposed to represent?
Discussion
L700: Change “more good quality insects” to something such as “higher quality insects”.
Author Response
Please see the attachment. Responses are in Red font.
